# Cryo-EM reveals distinct conformations of *E. coli* ATP synthase on exposure to ATP

Meghna Sobti[1,2], Robert Ishmukhametov[3], James C Bouwer[4], Anita Ayer[2,5], Cacang Suarna[5], Nicola J Smith[2,6], Mary Christie[1,2†], Roland Stocker[2,5], Thomas M Duncan[7], Alastair G Stewart[1,2]*

[1]Molecular, Structural and Computational Biology Division, The Victor Chang Cardiac Research Institute, Darlinghurst, Australia; [2]St Vincent's Clinical School, Faculty of Medicine, UNSW Sydney, Sydney, Australia; [3]Department of Physics, Clarendon Laboratory, University of Oxford, Oxford, United Kingdom; [4]Molecular Horizons, The University of Wollongong, Wollongong, Australia; [5]Vascular Biology Division, Victor Chang Cardiac Research Institute, Darlinghurst, Australia; [6]Molecular Cardiology and Biophysics Division, Victor Chang Cardiac Research Institute, Darlinghurst, Australia; [7]Department of Biochemistry & Molecular Biology, SUNY Upstate Medical University, Syracuse, NY, United States

**Abstract** ATP synthase produces the majority of cellular energy in most cells. We have previously reported cryo-EM maps of autoinhibited *E. coli* ATP synthase imaged without addition of nucleotide (Sobti et al. 2016), indicating that the subunit ε engages the α, β and γ subunits to lock the enzyme and prevent functional rotation. Here we present multiple cryo-EM reconstructions of the enzyme frozen after the addition of MgATP to identify the changes that occur when this ε inhibition is removed. The maps generated show that, after exposure to MgATP, *E. coli* ATP synthase adopts a different conformation with a catalytic subunit changing conformation substantially and the ε C-terminal domain transitioning via an intermediate 'half-up' state to a condensed 'down' state. This work provides direct evidence for unique conformational states that occur in *E. coli* ATP synthase when ATP binding prevents the ε C-terminal domain from entering the inhibitory 'up' state.
DOI: https://doi.org/10.7554/eLife.43864.001

*For correspondence:
a.stewart@victorchang.edu.au

Present address: †School of Life and Environmental Sciences, University of Sydney, Sydney, Australia

Competing interests: The authors declare that no competing interests exist.

## Introduction

The majority of metabolic energy in cells is generated by $F_1F_o$ ATP synthase, a biological rotary motor that converts the proton motive force (pmf) to adenosine tri-phosphate (ATP) in both oxidative phosphorylation and photophosphorylation (*Stewart et al., 2014*; *Walker, 2013*). The enzyme consists of two reversible rotary motors, termed $F_1$ and $F_o$, coupled together by one central and one peripheral stalk, with the simplest subunit composition found in bacteria such as *Escherichia coli* (*Figure 1—figure supplement 1*). The $F_o$ motor spans the membrane and converts the potential energy of the pmf into rotation of the central stalk that in turn drives conformational changes in the three catalytic sites of the $\alpha_3\beta_3$ $F_1$ motor subunits to generate ATP. Moreover, this process is reversible so that, if the pmf drops below the threshold needed to power ATP synthesis, the motor has the ability to reverse and, in some bacterial species, operates primarily as a proton pump driven by ATP hydrolysis. Regulation of these ATPase/synthase activities is particularly important in times of cellular stress, primarily to prevent wasteful ATP consumption. However, different regulatory mechanisms are used by different $F_1F_o$ subtypes (bacterial, chloroplastic, mitochondrial) and species

(*Sielaff et al., 2018*; *Morales-Rios et al., 2015*; *Stewart and Stock, 2012*; *Gledhill et al., 2007*; *Hahn et al., 2018*; *Stewart, 2014*; *Pullman and Monroy, 1963*).

It has been postulated that bacterial ATP synthases are regulated by several mechanisms, with conformational changes of the ε subunit and catalytic nucleotide occupancies likely playing major roles (*Sielaff et al., 2018*; *Hyndman et al., 1994*; *Feniouk et al., 2006*). A long-standing question surrounds the role of the C-terminal domain of the ε subunit (εCTD). The εCTD is known to change conformation and block rotation of the central stalk in the $F_1$ motor from several bacteria, but this role is absent in the enzyme of mitochondria (*Sielaff et al., 2018*; *Laget and Smith, 1979*; *Cingolani and Duncan, 2011*; *Sobti et al., 2016*; *Yagi et al., 2007*; *Shirakihara et al., 2015*). In the active bacterial enzyme, the εCTD is proposed to adopt a condensed or 'down' conformation (*Krah et al., 2017*), and this state has been observed for isolated ε subunit from *E. coli* (*Uhlin et al., 1997*; *Wilkens and Capaldi, 1998a*) and *Geobacillus stearothermophilus* (or *Bacillus* PS3, hereafter termed PS3) (*Yagi et al., 2007*) as well as in isolated $F_1$ from *Caldalkalibacillus thermarum* (*Ferguson et al., 2016*). In autoinhibited states observed in crystal structures of $F_1$ from *E. coli* and PS3, the εCTDs are extended in similar 'up' conformations, so that a helix inserts into the $F_1$ central cavity and appears to block rotation of the complex by binding to both the central rotor subunit γ and several surrounding α and β subunits (*Sielaff et al., 2018*; *Cingolani and Duncan, 2011*; *Mendoza-Hoffmann et al., 2018*) (*Figure 1—figure supplement 2*). However, there is a key structural difference between these 'up' states: in *E. coli,* the εCTD consists of two helices (here termed εCTH1 and εCTH2; see *Figure 1—figure supplement 2*) separated by an extended loop, with each helix interacting with a different region of the γ subunit (*Cingolani and Duncan, 2011*; *Rodgers and Wilce, 2000*); whereas in PS3, εCTH1 and εCTH2 instead join to form one continuous helix (*Shirakihara et al., 2015*). In PS3 and a related *Bacillus* species, ATP is known to bind to the ε subunit and stabilise the 'down' conformation (*Yagi et al., 2007*; *Krah et al., 2017*; *Kato-Yamada and Yoshida, 2003*; *Kato-Yamada, 2005*; *Imamura et al., 2009*), and a clear mechanism of regulation by the εCTD and ATP levels has been proposed (*Shirakihara et al., 2015*). However, the physiological importance of the εCTD in ATP synthases of other bacteria, such as *E. coli*, has remained uncertain. Crosslinking studies have suggested that the εCTD of *E. coli* ATP synthase could act as a ratchet, preventing hydrolysis - but not synthesis - of ATP (*Tsunoda et al., 2001*). However, deletion studies performed in vivo, in which the *E. coli* εCTD was removed, showed minimal impact on cell growth or viability (*Shah and Duncan, 2015*), although deletion of just the five terminal residues results in decreased respiratory growth due to increased inhibition of ATP synthesis (*Taniguchi et al., 2011*). For the highly latent enzyme from *C. thermarum*, an early study indicated the εCTD is involved in inhibition (*Keis et al., 2006*), but thus far crystallographic studies of its $F_1$ complex have shown the εCTD in the down conformation irrespective of ATP binding to ε (*Ferguson et al., 2016*). Despite confusion over the possible importance of the εCTD for the bacteria noted above, one recent study demonstrated that the εCTD can impact the virulence of *Streptococcus pneumoniae*. $F_1F_o$ is essential for viability of *S. pneumoniae* and functions in pH homeostasis and in its acid tolerance response (*Ferrándiz and de la Campa, 2002*; *Cortes et al., 2015*). In a mouse model of pneumococcal bacteraemia, a frameshift mutation that scrambled the sequence of the εCTD was found to increase pneumococcal survival in the spleen, most likely by reducing killing by splenic macrophages (*Gerlini et al., 2014*). Therefore, it is important to characterize more fully the bacterial εCTD and its different conformational states and interactions within bacterial ATP synthases.

In contrast to the crystallographic reports mentioned above with isolated ε subunits or soluble $F_1$, our previous cryo-EM study (*Sobti et al., 2016*) provided maps of the intact *E. coli* ATP synthase. These data were obtained using purified protein without the addition of exogenous nucleotide, thus likely representing a state autoinhibited by ε, here termed ATP synthase[AI]. Here we have used similar methodologies to investigate the effect of ATP on the conformation of *E. coli* $F_1F_o$ ATP synthase, by cryo-freezing and subsequently imaging the enzyme after a brief incubation with 10 mM MgATP. The maps obtained (termed ATP synthase[+ATP] hereafter) indicate that, in the presence of MgATP, the εCTD became detached from the body of the enzyme, coupled with a large change in one of the three catalytic β subunits. Further analysis identified an intermediate in which the εCTD was in a

'half-up' conformation, suggesting that the enzyme can still rotate, even when the ε subunit is not in its condensed down position, as well as an εCTD 'down' conformation likely showing an active intermediate. Further inspection of our maps demonstrated peaks present in all six nucleotide-binding sites in the ATP synthase[+ATP] enzyme (three catalytic β sites and three noncatalytic α sites), in contrast to the autoinhibited protein in which only four binding pockets (a single catalytic β site and three noncatalytic α sites) were occupied with nucleotide. Together with our previous work, these results reveal three distinct positions taken up by the εCTD in response to different catalytic nucleotide–bound states, providing further insight into the mechanism of ATP synthase regulation in *E. coli* and related species.

## Results

### Structure of *E. coli* $F_1F_o$ ATP synthase in the presence of ATP

To determine the structure of the *E. coli* $F_1F_o$ ATP synthase in the presence of ATP, the detergent-solubilized *E. coli* enzyme (*Sobti et al., 2016*) was incubated with 10 mM ATP (containing 50 μM ADP [n = 3; stdev = 2 μM]) and 10 mM $MgCl_2$ for 30 s at 20°C prior to preparing grids for cryo-EM. Sample preparation took a further 15 s before cryogenic temperature was reached, with approximately 0.25 mM ATP hydrolysed to ADP in the total time for preparation (45 s) (*Figure 1—figure supplement 3*). The time used for these experiments was optimized to be within the linear range of ADP production, so that uninhibited ATP synthase could be observed, with an ADP concentration of ~0.3 mM at the time of freezing. 10 mM ATP and a 45 s preparation time were chosen to emulate the concentrations seen in *E. coli* growing under aerobic conditions, where 9.6 mM ATP and 0.6 mM ADP has been observed (*Yaginuma et al., 2014*; *Bennett et al., 2009*).

The grids produced after incubation with ATP were similar to those used in our previous study of the autoinhibited enzyme and micrographs showed clear ATP synthase particles. A dataset of 8509 movie micrographs was collected at 200 kV accelerating voltage, of which 7858 were selected based on image quality. 579,942 particles were picked and 319,315 of these appeared to be of the intact complex (*Figure 1—figure supplement 4*). Three different rotational conformations could be filtered from these data (each containing 97,095, 72,757 and 51,534 particles) that were related by a 120° rotation of the central stalk, displaying the hallmark dynamic movements predicted for ATPases/synthases (*Stewart et al., 2012*). The quality of the maps obtained for each conformation was comparable, with resolutions of 5.0 Å, 5.3 Å and 5.5 Å (*Figure 1a* and *Figure 1—figure supplements 5* and *6*). However, the proportion of particles corresponding to each reconstruction differed from our previous study (33% State 1, 44% State 2 and 23% State 3 in this study, compared with 46% State 1, 30% State 2 and 24% State 3 in *Sobti et al., 2016*), suggesting that a larger number of molecules were now observed in State 2 rather than State 1. Single molecule studies have also observed the enzyme populating the rotational dwell states in different proportions (*Sielaff et al., 2019*), suggesting that each dwell state may be at a different energy minimum. Inspection of the nucleotide-binding pockets in the α and β subunits revealed two further peaks, in addition to the four nucleotide peaks we reported previously for the autoinhibited enzyme (*Sobti et al., 2016*) (*Figure 1—figure supplement 7*). This indicated that all six nucleotide-binding sites were occupied, even though one β subunit ($β_E$) remained in the open, low-affinity conformation (*Figure 1—figure supplement 1b*).

Re-refinement of the ATP synthase[AI] maps presented in our previous study (*Sobti et al., 2016*) improved their nominal resolution to 5.7 Å, 6.6 Å and 7.2 Å (previously 6.9 Å, 7.8 Å and 8.5 Å, respectively), and enabled features to be seen more clearly (*Figure 1b* and *Figure 1—figure supplement 8*). The re-refined maps were used to compare ATP synthase before and after incubation with ATP. Compared with ATP synthase[AI], the ATP synthase[+ATP] reconstructions showed a large conformational change in one of the three β subunits ($β_{DP}$) and that the εCTD was absent from the maps (*Figure 1*). The conformation of the catalytic subunits in the ATP synthase[+ATP] map closely resembles that of the isolated $F_1$ enzyme from *C. thermarum* (*Ferguson et al., 2016*) (*Figure 1—figure supplement 9*), except that no density was detectable for the εCTD. Even inspecting the unmasked unfiltered half maps at low threshold, it was difficult to formulate a clear idea of the position of the εCTD, suggesting that under these conditions it may be sampling multiple conformations.

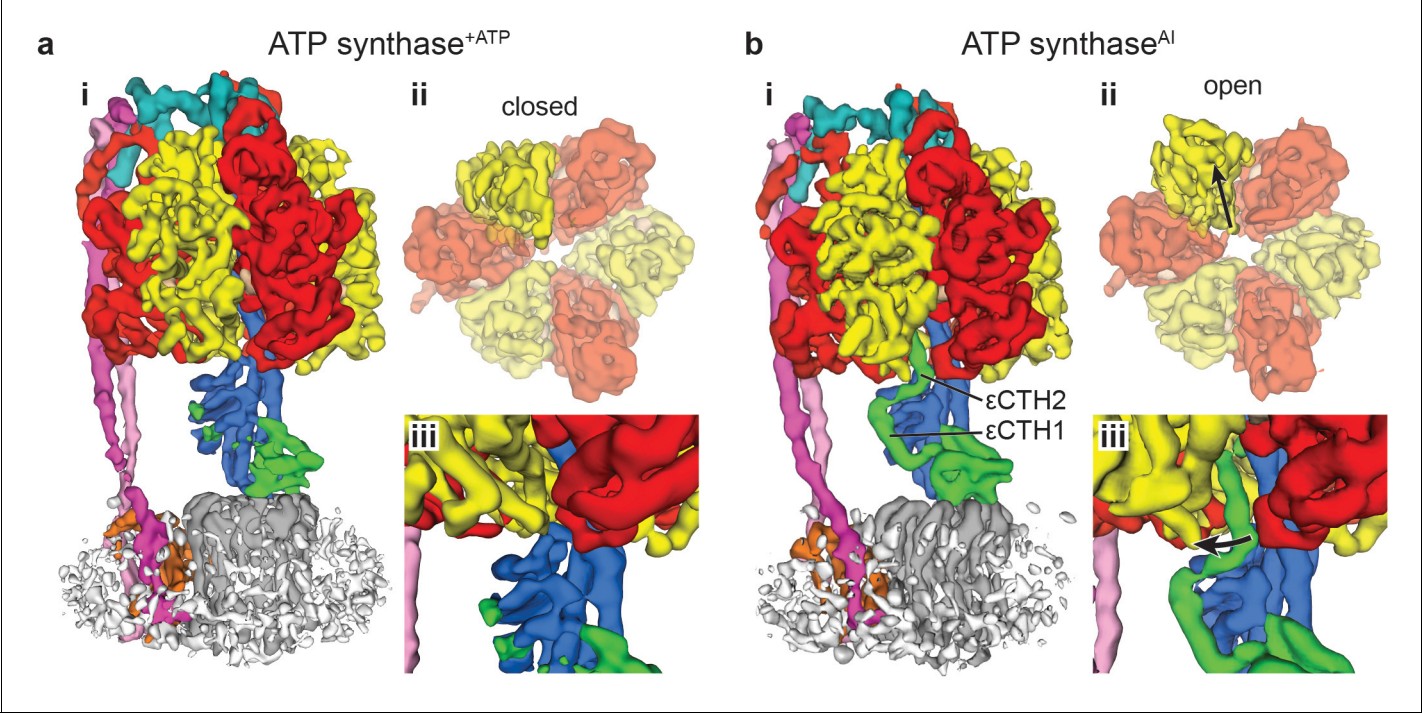

**Figure 1.** Comparison of the maps obtained (**a**) in the presence of ATP (ATP synthase[+ATP]) and (**b**) the autoinhibited state in the absence of ATP (ATP synthase[AI]). Comparison of the active and autoinhibited maps shows that: (**i**) the overall architecture is similar; (**ii**) a single β subunit (brighter yellow) is in a closed conformation in the active form and open conformation in the autoinhibited form (black arrows show the direction of movement – view is of the segmented α and β subunits, viewed from below); (**iii**) density corresponding to the εCTD (coloured green) is very weak in the ATP synthase[+ATP] map, suggesting it has multiple conformations, whereas in the ATP synthase[AI] map, the εCTD is clearly identifiable in the 'up' position, with the εCTH1 and εCTH2 bound to the γ subunit (coloured blue).

DOI: https://doi.org/10.7554/eLife.43864.002

The following figure supplements are available for figure 1:

**Figure supplement 1.** Schematic of *E. coli* $F_1F_o$ ATP synthase.
DOI: https://doi.org/10.7554/eLife.43864.003

**Figure supplement 2.** Autoinhibited structures of bacterial $F_1$-ATPases.
DOI: https://doi.org/10.7554/eLife.43864.004

**Figure supplement 3.** *E. coli* ATP synthase hydrolyses ATP to ADP.
DOI: https://doi.org/10.7554/eLife.43864.005

**Figure supplement 4.** Representative micrograph and top ten 2D classifications.
DOI: https://doi.org/10.7554/eLife.43864.006

**Figure supplement 5.** Three rotational conformations of the *E. coli* $F_1F_o$ ATP synthase after incubation with ATP.
DOI: https://doi.org/10.7554/eLife.43864.007

**Figure supplement 6.** Flowchart describing cryo-EM data analysis.
DOI: https://doi.org/10.7554/eLife.43864.008

**Figure supplement 7.** Nucleotide occupancy of the ATP synthase[AI] and ATP synthase[+ATP] maps.
DOI: https://doi.org/10.7554/eLife.43864.009

**Figure supplement 8.** Three rotational conformations of the *E. coli* $F_1F_o$ ATP synthase in their autoinhibited form.
DOI: https://doi.org/10.7554/eLife.43864.010

**Figure supplement 9.** The crystal structure of *C.thermarum* $F_1$ ATPase compared to the ATP synthase[+ATP] cryo-EM map.
DOI: https://doi.org/10.7554/eLife.43864.011

**Figure supplement 10.** Position of the cysteines in wild-type *E. coli* $F_1$-ATPase.
DOI: https://doi.org/10.7554/eLife.43864.012

**Figure supplement 11.** Fit of *C. thermarum* ε subunit into the εCTD 'down' ATP synthase[+ATP] cryo-EM map, highlights density corresponding to the ATP binding site.
DOI: https://doi.org/10.7554/eLife.43864.013

**Figure supplement 12.** ATPase activity of digitonin solubilized *E. coli* $F_1F_o$ ATP synthase.
DOI: https://doi.org/10.7554/eLife.43864.014

To identify the position of the εCTD, the data were processed more extensively and refined into sub-classifications (*Figure 1—figure supplement 6*). Although the resolution obtained in these sub-classes was lower than in the general classification and they displayed the same rotational states, they showed additional weak density corresponding to the εCTD in either a condensed 'down' position (*Figure 2a* and *Figure 2—figure supplement 1a*), or intermediate 'half-up' position (*Figure 2b* and *Figure 2—figure supplement 1b*). In the 'down' position, weak density was present for both εCTH1 and εCTH2 with the helices arranged adjacent to one another, attached to the N-terminal region of the ε subunit. However, in the intermediate 'half-up' position weak density was only observed for the εCTH1 bound to the γ subunit, with no identifiable density for the εCTH2, suggesting that this helix was mobile.

## Peripheral stalk structure

The peripheral stalk of ATP synthase is constructed from a right-handed coiled coil and holds the catalytic α and β subunits stationary relative to a rotating central stalk (*Wilkens and Capaldi, 1998b*; *Lee et al., 2010*; *Walker and Dickson, 2006*). Our previous maps (*Sobti et al., 2016*) described the overall peripheral stalk architecture in *E. coli* $F_1F_o$ ATP synthase, to which density was attributed as belonging to two b subunits and one δ subunit. In light of recent higher resolution cryo-EM maps of the eukaryotic $F_1F_o$ ATP synthase from *Spinacia oleracea* chloroplasts (*Hahn et al., 2018*), we attempted to assign the density of the peripheral stalk more precisely in the new maps. Of particular interest was a small density peak (*Figure 3a–c*, pink density) that cannot be identified in other structures of related ATP synthases (*Morales-Rios et al., 2015*; *Hahn et al., 2018*). However, this density can be identified in all cryo-EM maps produced of *E. coli* $F_1F_o$ ATP synthase thus far (EMD-8357, EMD-8358, EMD-8359 and the maps presented in this study). The length of the density relating to

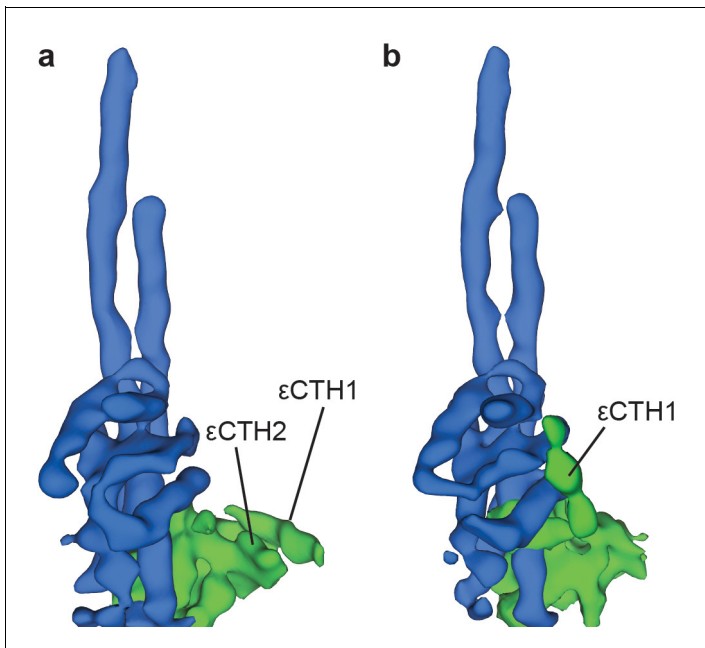

**Figure 2.** Two conformations of subunit ε are observed in sub-classification of ATP synthase$^{+ATP}$ maps. Processing and refinement of the ATP synthase$^{+ATP}$ dataset using Relion (*Scheres, 2012*) identified two conformations for the εCTD. These show a 'down' conformation (**a**) where the εCTH1 and εCTH2 are bound to the N-terminal region of the ε subunit and a 'half-up' conformation (**b**) where the εCTH1 was still bound to the γ subunit but the εCTH2 was not visible even at low thresholds.

DOI: https://doi.org/10.7554/eLife.43864.015

The following figure supplement is available for figure 2:

**Figure supplement 1.** Two cryo-EM maps filtered from the ATP synthase$^{+ATP}$ dataset show conformational changes in subunit ε.

DOI: https://doi.org/10.7554/eLife.43864.016

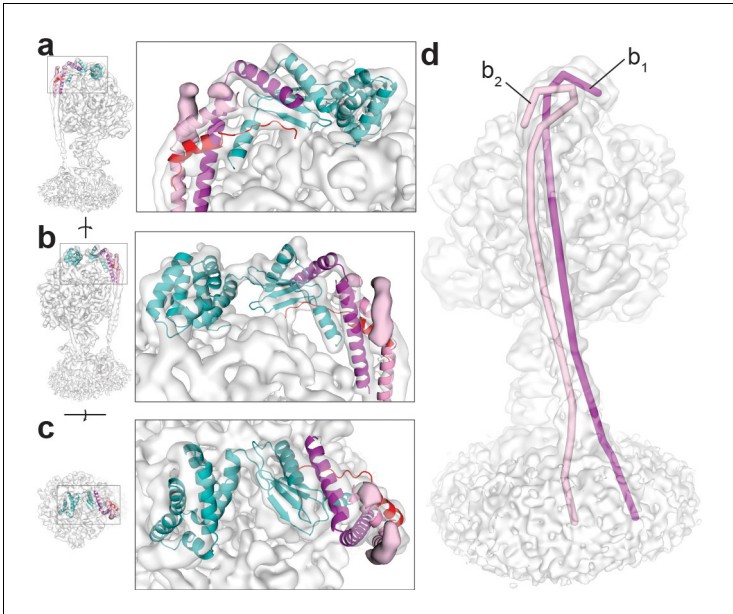

**Figure 3.** Density observed for the $F_1F_o$ ATP synthase peripheral stalk. The cryo-EM map of ATP synthase[+ATP] docked with the atomic model of *S. oleracea* chloroplast ATP synthase peripheral stalk (*Hahn et al., 2018*), reveal a region of difference density (coloured pink density in **a**), (**b and c**) not seen in related F-ATP synthases. (**a and b**) side views and (**c**) top view. Coot (*Emsley et al., 2010*) and PyMol (Schrödinger) were used to create a line object that traced the density believed to be the b subunits. The magenta line (labelled $b_1$) measured 233 Å and the pink line (labelled $b_2$) measured 227 Å (**d**), suggesting that the difference density seen corresponds to the C-terminus of a single $b_2$ subunit. The $b_1$ and $b_2$ subunits of the *E. coli* ATP synthase peripheral stalk follow different trajectories across the $F_1$ motor surface.

DOI: https://doi.org/10.7554/eLife.43864.017

The following figure supplements are available for figure 3:

**Figure supplement 1.** Comparison of the membrane anchoring domain.
DOI: https://doi.org/10.7554/eLife.43864.018

**Figure supplement 2.** Comparison of peripheral stalk conformations in the three rotational states.
DOI: https://doi.org/10.7554/eLife.43864.019

---

the b subunits was measured to investigate whether this additional density could be attributed to the b subunits, and found to be 233 Å and 227 Å (termed $b_1$ and $b_2$, respectively; *Figure 3d*). Together with the prediction that residues 66–122 of the b subunits form an unstaggered coiled coil in the isolated dimer (*Del Rizzo et al., 2002*), this suggests that the additional density peak represents the C-terminus of just the $b_2$ subunit. The differing conformation between the $b_1$ and $b_2$ subunits seen here appears to be unique to some bacterial ATP synthases and is likely due to their being a homodimer in these systems. Other related F-ATP synthases, such as *Paracoccus denitrificans*, contain two different b subunits (termed b and b') that differ in length and sequence (*Morales-Rios et al., 2015*) and therefore can adopt their own individual folds. In *E. coli* ATP synthase, each b subunit is required to adopt two different conformations using the same amino acid sequence. This may be due to the requirement of the homodimer binding to asymmetric α and δ surfaces, whereby the $b_1$ subunit forms the majority of the contacts between the $F_1$ motor while the $b_2$ subunit folds back onto itself providing rigidity to the peripheral stalk. Most recently, a high-resolution structure of PS3 ATP synthase has been reported using cryo-EM (*Guo and Rubinstein, 2018*). Although the maps and models are not currently available, similar density can be seen suggesting that the break in symmetry of the b subunits observed at the top of the $F_1$ motor is conserved across species which contain a homodimeric b subunit. Moreover, analysis of our improved maps of the *E. coli* ATP synthase[AI] (particularly State 2, which shows the clearest density in the $F_o$ region) corroborates the results from the far higher resolution structure of the PS3 enzyme, showing a similar overall fold in the membrane $F_o$ region to spinach chloroplast ATP synthase (*Hahn et al., 2018*), although following

slightly different trajectories (*Figure 3—figure supplement 1*). However, because the maps of the *E. coli* enzyme show only weak density in this region it is difficult to draw a firm conclusion on this point.

## Discussion

The structural information obtained here using *E. coli* $F_1F_o$ ATP synthase incubated with MgATP likely reflects conformations uninhibited by the εCTD. Although the maps generated are of limited resolution, the position of the catalytic subunits and εCTD were clearly identifiable. Two different εCTD conformations were observed: a 'down' conformation, in which εCTH1 and εCTH2 are arranged next to one another and interacts with the N-terminal region of the ε subunit, and a 'half-up' conformation in which the εCTH1 lies against the γ subunit (*Figure 2*). Because the ATP and ADP concentrations present were comparable to those observed in intact cells (*Bennett et al., 2009*) it is likely that both these conformations are present in *E. coli*. Furthermore, analysis of the maps in light of the recent high-resolution structure of related *S. oleracea* enzyme has highlighted a unique C-terminal fold of the peripheral stalk b subunits in *E. coli* ATP synthase that derives from the homodimeric nature of this subunit in this system.

A considerable body of work has described the inhibition of $F_1$-ATPases by ADP, $Mg^{2+}$ and εCTD (*Sielaff et al., 2018*; *Hyndman et al., 1994*; *Laget and Smith, 1979*; *Cingolani and Duncan, 2011*; *Sobti et al., 2016*; *Richter, 2004*; *Zhou et al., 1988*; *Avron, 1962*; *Drobinskaya et al., 1985*; *Guerrero et al., 1990*). Because the sample in the present study was imaged under conditions in which ADP accumulated at a linear rate (*Figure 1—figure supplement 3*), we propose that the maps represent structures of *E. coli* $F_1F_o$ ATP synthase in which the autoinhibitory state of the εCTD is prevented. These results indicate that ATP induces the εCTD to exit from the central cavity of $F_1$ and so facilitate rotation of the enzyme. The contribution of the εCTD to regulate *E. coli* $F_1F_o$ ATP synthase activity has previously been unclear. Our structural studies on *E. coli* $F_1F_o$ ATP synthase support a mechanism whereby the εCTD is primed to act as a 'safety lock" (*Feniouk and Junge, 2005*) or 'ratchet" (*Tsunoda et al., 2001*; *Cipriano and Dunn, 2006*) to prevent ATP synthase from entering the ADP/$Mg^{2+}$ inhibited state (*Shah et al., 2013*) even when ADP and $Mg^{2+}$ concentrations are high (*Figure 4*). The 'half-up' state that we identified in our cryo-EM maps has been previously hypothesised (*Cingolani and Duncan, 2011*) and studies on the enzyme in the absence of the εCTH2 support the hypothesis that the εCTD 'half-up' conformation can act as a weak inhibitor, partially inhibiting without insertion into the $F_1$ motor (*Nakanishi-Matsui et al., 2014*). It is unlikely that the 'half-up' conformation we observe here is an artefact generated by the cysteine residue free construct used in this study, because in our previous work (*Sobti et al., 2016*) the same cysteine-free enzyme did not show the 'half-up' state. In addition, previous studies using γC87S mutant protein did not show any change in ε inhibition (*Duncan et al., 1995*), and in the structure of the native enzyme (*Figure 1—figure supplement 10*) loops in the γ subunit separate the cysteines in subunit γ and residues in the εCTH1, so that they are not in direct contact.

Structural data from this and previous studies suggest that loading multiple catalytic sites with nucleotides disrupts access of the εCTD to the inhibitory 'up' state in *E. coli* (*Figure 4*). In a nucleotide-free solution, the $F_1$-ATPase adopts an autoinhibited state (seen in the crystal structure of isolated *E. coli* $F_1$-ATPase (*Cingolani and Duncan, 2011*) and the ATP synthase$^{AI}$ cryo-EM map (*Sobti et al., 2016*); *Figure 4a and b*). The isolated $F_1$-ATPase shows one β subunit ($β_{DP}$) in a 'half-closed' state, with ADP bound, which interacts directly with the 'up' conformation of the εCTH2 (*Figure 4a*). In the ATP synthase$^{AI}$ cryo-EM map, the same β subunit is unoccupied and has adopted an open conformation that no longer interacts with the εCTH2, which could facilitate a reversible mechanism whereby the helix is able to escape from its inserted position. When compared with isolated $F_1$, the reduced interactions with the 'up' state of subunit ε seen in the complete $F_1F_o$ complex may explain why ε inhibition is significantly reduced upon rebinding of *E. coli* $F_1$ to $F_o$. In the ATP synthase$^{+ATP}$ cryo-EM maps, all catalytic sites are at least partially occupied with nucleotide (*Figure 1—figure supplement 7*) and the εCTD is seen in two conformations that no longer contact the α and β subunits (*Figure 4c and d*). Because we only observed the εCTD in the 'up' conformation in the absence of ATP, these results indicate that nucleotide binding to the catalytic subunits, in either one or a combination of catalytic sites, is important for the release of the εCTD. Binding of ATP is unlikely to accelerate release of the εCTD from the 'up' state, as that would conflict with the known

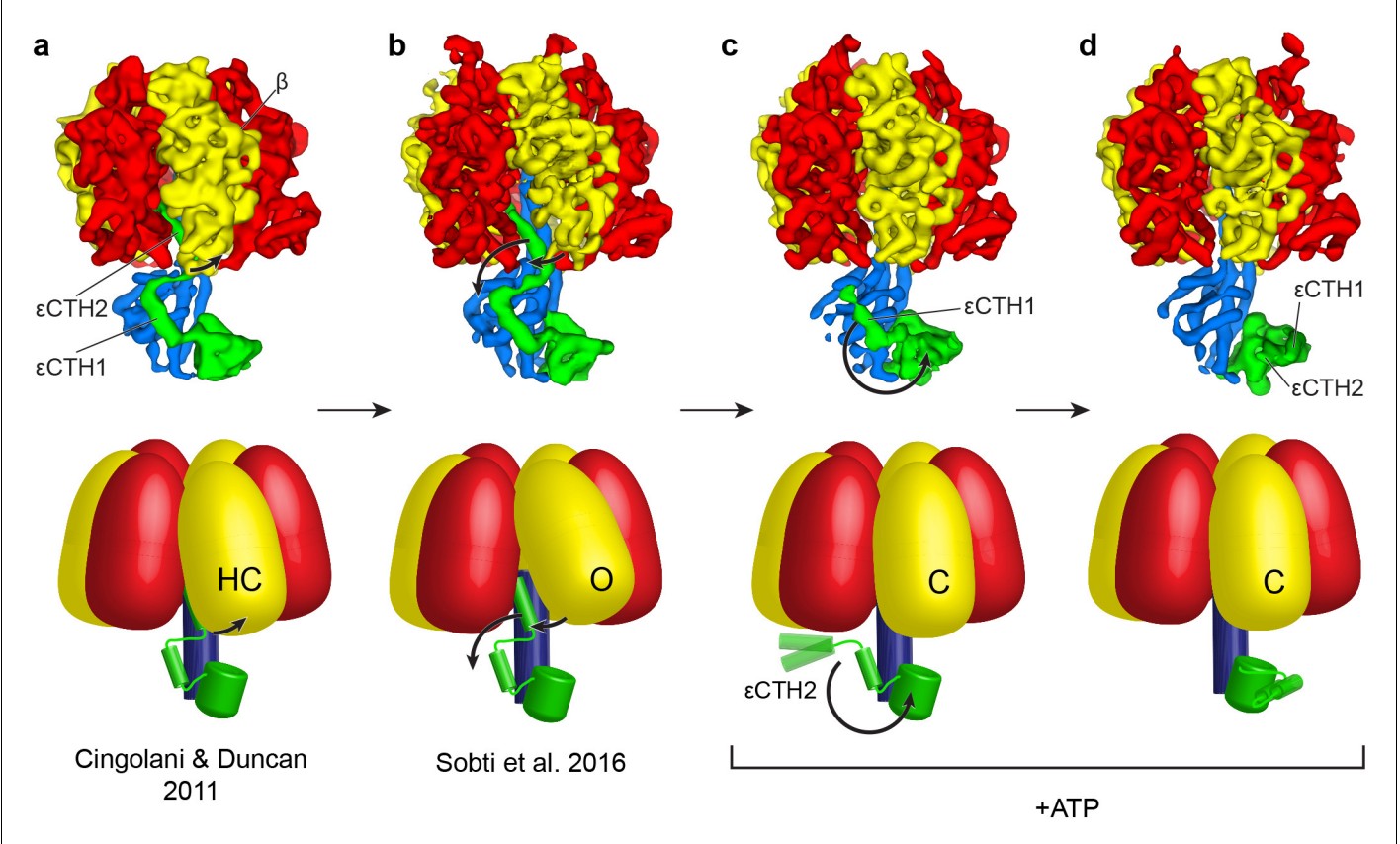

**Figure 4.** Proposed mechanism for the release of εCTD from the inhibitory 'up' conformation in *E. coli* F$_1$-ATPase. Structural models presented as maps (above; *E. coli* crystal structure in (a) presented as low smooth map for comparison) and simple schematics (below) to describe the possible structural movements of *E. coli* F$_1$-ATPase during activation. In a nucleotide-free solution, the F$_1$-ATPase adopts an inactive state (seen in the crystal structure of isolated *E. coli* F$_1$-ATPase (*Cingolani and Duncan, 2011*) and the ATP synthase[AI] cryo-EM map (*Sobti et al., 2016*; (a) and (b) respectively). The isolated F$_1$-ATPase shows one of the β subunits in a 'half-closed' state (labelled as HC), interacting with the 'up' conformation of the εCTH2 (a). In the cryo-EM maps of the intact enzyme (b), the same β subunit adopts an open conformation (labelled as O) where it no longer interacts with the εCTH2, allowing the reversible release of the helix from its inserted position. In the presence of ATP, the εCTD is seen in two conformations (c and d) that no longer contact the α and β subunits, and the aforementioned β subunit then adopts a closed state (labelled as C) that prevents the εCTD from returning to the 'up' conformation. The εCTD can either be in a 'half-up' (c) conformation, with the εCTH2 mobile, or a 'down' conformation (d) with both εCTH1 and εCTH2 in a condensed form.

DOI: https://doi.org/10.7554/eLife.43864.020

noncompetitive nature of ε inhibition with *E. coli* F$_1$ and F$_o$F$_1$. Rather, as the εCTH2 escapes from the central cavity of the ATP synthase[AI] conformation (*Figure 4b*), rapid binding of ATP would induce a transition from the open (*Figure 4b*, 'O') to the closed conformation (*Figure 4c/d*, 'C'), preventing re-entry of εCTH2 until the next catalytic dwell position. Finally, the 'half-up' conformation seen for the εCTD supports the prediction that the εCTH2 can remain mobile, anchored to γ by the εCTH1, so that it can re-enter the central cavity at a later step in the cycle (*Cingolani and Duncan, 2011*). It is also possible that this εCTH1 interaction alone may be able to modulate enzyme activity, as seen in a previous deletion study (*Nakanishi-Matsui et al., 2014*).

The 'down' conformation of the ε subunit has been observed for the isolated ε subunit of PS3 with ATP bound (*Yagi et al., 2007*) and on F$_1$ from *C. thermarum* (*Ferguson et al., 2016*). For ATP synthase[+ATP] cryo-EM maps containing the εCTD in the 'down' conformation, close inspection showed very weak density, consistent with a fraction of these complexes showing nucleotide bound to the ε subunit. Docking of the *C. thermarum* (*Ferguson et al., 2016*) ε subunit with ATP bound, showed good correlation between the position occupied by the nucleotide and additional density seen in the ATP synthase[+ATP] map (*Figure 1—figure supplement 11*). Partial occupancy would be

consistent with the concentration of ATP employed in our study being slightly below the $K_d$ of ~22 mM observed for ATP binding to isolated *E. coli* ε subunit (*Yagi et al., 2007*), and so because weak density attributable to nucleotide bound to the εCTD was only seen with the down conformation, it is possible that the 'half-up' conformation, with the associated probable disorder of εCTH2, might result when nucleotide is not bound to the ε subunit. Further work, for example, using a range of ATP concentrations, will be needed to resolve this point.

Although the movements of the εCTD are well defined by this study, the resolution of the reconstructions is limited compared to others (*Guo and Rubinstein, 2018*; *Murphy et al., 2019*). We believe this is likely due to inherent flexibility in the complex, possible varying subunit or lipid stoichiometry (*Chorev et al., 2018*), the detergent used, ice thickness, particle density and the lower accelerator voltage used during image acquisition (200 vs. 300 kV). Moreover, we only observed three rotational states whereas seven were seen in bovine mitochondrial ATP synthase (*Zhou et al., 2015*) and thirteen in *Polytomella sp.* mitochondrial ATP synthase (*Murphy et al., 2019*). We did employ methods similar to those used in these studies in an attempt to observe rotary sub-steps in the *E. coli* enzyme, but were unsuccessful. Although the complex should be undergoing rotation, due to the ATP hydrolysis that is occurring during freezing, single molecule studies have shown that in order for the complex to spend significant time in rotary sub-steps it had to be imaged under high drag, such as in 30% polyethylene glycol with a large object attached (*Ishmukhametov et al., 2010*). Indeed, the poor density seen at the membrane interface of the subunit b dimer, points towards these reconstructions containing multiple conformations, which may contribute to limiting the resolution obtained. It is possible that further work to obtain a much larger dataset, containing many more particles, may enable identification of more of the rotary sub-steps that are likely present in this sample.

Recent work on *Polytomella sp.* mitochondrial ATP synthase (*Murphy et al., 2019*) has described this ATP synthase in great detail, showing a flexible coupling between the $F_1$ and $F_o$ motors mediated by a hinge region in *OSCP* (analogous to the δ subunit in *E. coli* ATP synthase). Comparison of the peripheral stalks in this study, by superposition of either the $F_o$ stator or the N-terminal region of subunit δ (*Figure 3—figure supplement 2*), shows that the peripheral stalk likely bends and twists as the enzyme undergoes rotary catalysis, as suggested in previous studies (*Hahn et al., 2018*; *Stewart et al., 2012*). We do not observe movement in the delta/OSCP hinge, most likely due to our inability to resolve rotary substates in our maps.

In previous studies on *E. coli* ATP synthase, the enzyme has been shown to become inactive four hours after purification, possibly due to aggregation (*Ishmukhametov et al., 2010*). In this study, the enzyme was purified rapidly (see Materials and methods) and EM grids were made shortly after purification to ensure active enzyme was being imaged. The images obtained showing free monodisperse protein with little evidence of aggregation. Furthermore, the enzyme was purified in digitonin rather than the mixture of phosphatidylcholine, octyl glucopyranoside, sodium deoxycholate and sodium cholate used previously (*Ishmukhametov et al., 2010*; *Ishmukhametov et al., 2005*). Digitonin was selected because enzyme purified in the original mixture of lipids and detergents was not stable on the size exclusion chromatography (SEC) employed as a final purification step prior to grid freezing. Because the enzyme had been purified in a different detergent and further purified using SEC, we assessed the activity of the digitonin solubilized enzyme using ATP regeneration assays after 0, 1, 2, 3, 4 and 8 hr to compare with data taken on enzyme purified in previous studies (*Ishmukhametov et al., 2010*) (*Figure 1—figure supplement 12*). Importantly the enzyme showed little change in activity over 8 hr, suggesting that *E. coli* ATP synthase purified with digitonin and SEC may be less prone to inactivation or aggregation.

The present study has exploited the ability to manipulate sample conditions immediately before freezing to obtain images of *E. coli* ATP synthase at a time when ADP formation from ATP proceeds linearly, and hence likely unhindered, to identify the possible conformational changes that occur to remove the εCTD from the autoinhibitory position. This outcome was achieved by optimizing experimental design, considering particle concentration, as well as mixing and blotting time. Interestingly, employing similar methods on the related bacterial A/V type ATPase from *Thermus thermophilus*, showed that no conformational changes occur in the catalytic or related F subunits upon addition of the ATP transition state mimic fluoroaluminate (*Davies et al., 2017*). This could be due to different nucleotide affinity, operating temperature or indeed function of the complexes (*Nakano et al., 2008*).

In summary, this study presents structural data of *E. coli* ATP synthase imaged in the presence of ~9.75 mM ATP and ~0.3 mM ADP, comparable to the concentrations present in the intact bacterium. Compared to our previous work on the autoinhibited enzyme in the absence of ATP, the cryo-EM reconstructions presented here show that a single β subunit and the C-terminal domain of subunit ε change conformation and position substantially upon addition of ATP. Together, our studies provide strong support for the hypothesis that the ε subunit acts as an emergency brake to minimize wasteful hydrolysis of cellular ATP in situations where the ATP-to-ADP ratio is low. As the εCTD has recently been shown to impact pathogenesis of at least one bacterial species (*Gerlini et al., 2014*), the different conformations of the εCTD observed in these studies may have potential to guide future development of drugs that could target this bacteria-specific inhibitory mechanism.

## Materials and methods

ATP was purchased in the form of adenosine 5-triphosphate disodium salt hydrate (Sigma A7699).

### Protein purification

The same preparation was used as in *Sobti et al. (2016)*. However, the protein sample was concentrated to 11 µM (6 mg/ml) to improve particle density. An in-depth description of the methods used to isolate this protein is described below with approximate timings for each step.

Cysteine-free *E. coli* ATP synthase (with all cysteines residues substituted with alanine residues and a His-tag introduced on the β subunit) was expressed in *E. coli* DK8 strain (*Ishmukhametov et al., 2005*). Cells were grown at 37°C in LB medium supplemented with 100 µg/ml ampicillin for 5 hr. The cells were harvested by centrifugation at 5,000 g, obtaining ~1.25 g cells per litre of culture [~20 min for centrifugation step]. Cells were resuspended in lysis buffer containing 50 mM Tris/Cl pH 8.0, 100 mM NaCl, 5 mM MgCl$_2$, 0.1 mM EDTA, 2.5% glycerol and 1 µg/ml DNase I, and processed with three freeze thaw cycles followed by one pass through a continuous flow cell disruptor at 20 kPSI [~1 hr for lysis]. Cellular debris was removed by centrifuging at 7,700 × g for 15 mins, and the membranes were collected by ultracentrifugation at 100,000 × g for 1 hr [~1 hr 45 min for centrifugation steps]. The ATP synthase complex was extracted from membranes at 4°C for 1 hr by resuspending the pellet in extraction buffer consisting of 20 mM Tris/Cl, pH 8.0, 300 mM NaCl, 2 mM MgCl$_2$, 100 mM sucrose, 20 mM imidazole, 10% glycerol, 4 mM digitonin and EDTA-free protease inhibitor tablets (Roche) [~1.5 hr for resuspension]. Insoluble material was removed by ultracentrifugation at 100,000 g for 30 min. The complex was then purified by binding on Talon resin (Clontech) [~1 hr] and eluted in 150 mM imidazole [~1 hr for affinity step]. The protein was further purified with size exclusion chromatography on a 16/60 Superose six column equilibrated in a buffer containing 20 mM Tris/Cl pH 8.0, 100 mM NaCl, 1 mM digitonin and 2 mM MgCl$_2$ [~1 hr for gel filtration step]. The purified protein was then concentrated to 11 µM (6 mg/ml) [~15 min for concentration step], and snap frozen and stored for grid preparation. The total time from membranes to freezing was ~6 hr.

### ATPase enzymatic assay

The reaction mixture was started by vigorously mixing 27 µl of 11 µM (6 mg/ml) purified cysteine-free *E. coli* F$_1$F$_o$ ATP synthase and 3 µl of 100 mM ATP/100 mM MgCl$_2$ (pH 8.0) and incubated at 22°C. 5 µl of the reaction mixture was removed at 15, 30, 45, 60, 120 and 240 s time points and stopped immediately by adding 25 µl of ice-cold 0.4 M perchloric acid and mixed vigorously with a pipette. Samples were subsequently spun at 1700 x g at 4°C for 5 min. 30 µl of 1 M KH$_2$PO$_4$ and 7.5 µl of 3 M KOH was added to neutralize the mixture before a further spin at 1700 x g for 5 min at 4°C. The samples were then stored at −80°C prior to analysis via HPLC-UV.

### Analysis of ADP and ATP content by HPLC-UV

ADP and ATP in enzymatic reactions were detected as described previously in *Micheli et al. (1993)* with slight modifications. Briefly, 50 µl of sample was subjected to HPLC coupled with UV detection (Agilent 1200 series). ADP and ATP were separated on a Supelcosil C18 column (5 µM, 250 × 4.6 mm) by gradient elution using mobile phase A (0.1 M KH$_2$PO$_4$ pH 5.5 containing 6 mM tetrabutylammonium phosphate) and mobile phase B (100% methanol) at 1 mL/min. The gradient consisted of 7% mobile phase B (0–6 min) and 30% mobile phase B (6–11 min). The column was re-equilibrated

with 7% mobile phase B for 9 min. To obtain the ADP concentration in commercially bought ATP, a 10 mM ATP sample was loaded, and ADP and ATP were detected as described previously in *Huang et al. (2010)* and subtracted from subsequent readings. Briefly, ADP and ATP were separated on a Supelcosil C18 column (5 µM, 250 × 4.6 mm) by isocratic elution using 39 mM $KH_2PO_4$ pH 6.5 at 1 mL/min. For both chromatographic methods, ADP and ATP were detected by $UV_{254nm}$ and quantified by area comparison with authentic commercial standards (Sigma Aldrich, USA) and data analysed using ChemStation software (Agilent Technologies, USA).

## Cryo-EM grid preparation

1 µl of 100 mM ATP/100 mM $MgCl_2$ (pH 8.0) was added to an aliquot of 9 µl of purified cysteine-free *E. coli* $F_1F_o$ ATP synthase (*Sobti et al., 2016*; *Ishmukhametov et al., 2005*) at 11 µM (6 mg/ml) and the sample was incubated at 22°C for 30 s, before 3.5 µl was placed on glow-discharged holey gold grid (Ultrafoils R1.2/1.3, 200 Mesh). Grids were blotted for 3 s at 22°C, 100% humidity and flash-frozen in liquid ethane using a FEI Vitrobot Mark IV (total time for sample application, blotting and freezing was 15 s).

## Data collection

Grids were transferred to a Thermo Fisher Talos Arctica transmission electron microscope operating at 200 kV. Images were recorded automatically using EPU, yielding a pixel size of 0.98 Å. A total dose of 50 electrons per Å (*Walker, 2013*) was used, with the first 30 electrons spread over the initial 31 frames and the final 20 electrons captured as the final frame. The total exposure time was 62 s and the data were collected using a Falcon III in counting mode. 8509 movie micrographs were collected in two data collections (*Figure 1—figure supplement 6*).

## Data processing

### 200 kV ATP synthase[+ATP] dataset

MotionCorr2 (*Zheng et al., 2017*) was used to correct local beam-induced motion and to align resulting frames, with 5 × 5 patches. Defocus and astigmatism values were estimated using Gctf (*Zhang, 2016*) and 7858 micrographs were selected after exclusion based on ice contamination, drift and astigmatism. ~1000 particles from each data set were manually picked and subjected to 2D classification to generate templates for autopicking in RELION-3.0-beta (*Scheres, 2012*). The automatically picked micrographs were manually inspected to remove false positives, yielding 579,942 particles. These particles were re-extracted from motion corrected images that were dose-weighted and did not contain the final frame. These particles were then subjected to four rounds of 2D classification generating a final dataset of 319,315 particles. To reduce model bias, an independent initial map was made with RELION-3.0-beta using a subset of particles (13,461 particles) from the first data collection. Each dataset was classified into 3D classes using this initial model in RELION-3.0-beta, and similar classes were pooled yielding maps for ATP synthase[+ATP], related by a rotation of the central stalk (total: 221,386 particles; three states: 97,095, 72,757 and 51,534 particles). Subclassifictions were merged based on whether the εCTD could be observed in the 'half-up' or 'down' state (45,446 and 40,262 particles respectively). Final refinements were performed in cryoSPARC (*Punjani et al., 2017*). See *Figure 1—figure supplement 6* for a flowchart of the data processing strategy.

### 300 kV ATP synthase[AI] dataset

Particles from our previous study (*Sobti et al., 2016*), ATP synthase[AI], were refined using cryoSPARC (*Punjani et al., 2017*), resulting in superior maps and FSC.

### Map deposition

Maps for the ATP synthase[+ATP], εCTD 'half-down' and εCTD 'down' were deposited to the EMDB with accession codes EMD-9345, EMD-9346 and EMD-9348 respectively. Maps for the three ATP synthase[AI] were updated in the EMDB.

## Acknowledgments

AGS was supported by a National Health and Medical Research Council Fellowship APP1159347 and Grant APP1146403. RI was supported by BBSRC grant BB/L01985X/1. NJS was supported by a National Heart Foundation Future Leader Fellowship. MC was supported by Australian Research Council Fellowship DE160100608. We thank and acknowledge Dr Jessica Chaston, VCCRI, Mr Cameron Ding-Farrington, VCCRI, and Ms Caitlin Lawrence, the University of New South Wales, who aided in particle picking; Dr James Walshe, who aided in data processing; Dr Hari Venugopal, Monash Ramaciotti Centre for Cryo-Electron Microscopy, who helped with initial screening of samples. We would like to thank the use of the Victor Chang Innovation Centre, funded by the NSW Government, and the Electron Microscope Unit at UNSW, funded in part by the NSW Government. We wish to thank and acknowledge the use of the University of Wollongong Cryogenic Electron Microscopy Facility at Molecular Horizons under the Directorship of Dr. Antoine van Oijen.

## Additional information

### Funding

| Funder | Grant reference number | Author |
|---|---|---|
| National Health and Medical Research Council | APP1146403 | Meghna Sobti<br>Alastair G Stewart |
| Biotechnology and Biological Sciences Research Council | BB/L01985X/1 | Robert Ishmukhametov |
| Australian Research Council | DE160100608 | Mary Christie |
| National Health and Medical Research Council | APP1159347 | Alastair G Stewart |
| National Heart Foundation of Australia | Future Leader Fellowship | Nicola J Smith |

The funders had no role in study design, data collection and interpretation, or the decision to submit the work for publication.

### Author contributions

Meghna Sobti, Conceptualization, Formal analysis, Investigation, Methodology, Writing—original draft, Writing—review and editing; Robert Ishmukhametov, Conceptualization, Methodology, Writing—original draft, Writing—review and editing; James C Bouwer, Investigation, Methodology, Writing—review and editing; Anita Ayer, Formal analysis, Investigation, Methodology, Writing—review and editing; Cacang Suarna, Formal analysis, Investigation, Methodology; Nicola J Smith, Resources, Funding acquisition, Writing—review and editing; Mary Christie, Resources, Funding acquisition, Writing—original draft, Writing—review and editing; Roland Stocker, Resources, Funding acquisition, Investigation, Methodology, Writing—review and editing; Thomas M Duncan, Writing—original draft, Writing—review and editing; Alastair G Stewart, Conceptualization, Resources, Funding acquisition, Investigation, Methodology, Writing—original draft, Project administration, Writing—review and editing

### Author ORCIDs

Alastair G Stewart (iD) http://orcid.org/0000-0002-2070-6030

### Decision letter and Author response

Decision letter https://doi.org/10.7554/eLife.43864.036
Author response https://doi.org/10.7554/eLife.43864.037

## Additional files

### Supplementary files

- Transparent reporting form

DOI: https://doi.org/10.7554/eLife.43864.021

### Data availability

Cryo-EM maps have been deposited to the EMDB under accession numbers EMD-9345, EMD-9346, EMD-9348, EMD-20006, EMD-20007 and EMD-20008.

The following datasets were generated:

| Author(s) | Year | Dataset title | Dataset URL | Database and Identifier |
|---|---|---|---|---|
| Meghna Sobti, Alastair G Stewart | 2019 | Autoinhibited E. coli ATP synthase state 1 | http://www.ebi.ac.uk/pdbe/entry/emdb/EMD-20006 | Electron Microscopy Data Bank, EMD-20006 |
| Meghna Sobti, Alastair G Stewart | 2019 | Autoinhibited E. coli ATP synthase state 2 | http://www.ebi.ac.uk/pdbe/entry/emdb/EMD-20007 | Electron Microscopy Data Bank, EMD-20007 |
| Meghna Sobti, Alastair G Stewart | 2019 | Autoinhibited E. coli ATP synthase state 3 | http://www.ebi.ac.uk/pdbe/entry/emdb/EMD-20008 | Electron Microscopy Data Bank, EMD-20008 |
| Sobti M, Stewart AG, Sobti M, Alastair G Stewart | 2019 | E. coli ATP synthase after incubation with ATP - State C | http://www.ebi.ac.uk/pdbe/entry/emdb/EMD-9348 | Electron Microscopy Data Bank, EMD-9348 |
| Sobti M, Stewart AG | 2019 | E. coli ATP synthase after incubation with ATP - State A | http://www.ebi.ac.uk/pdbe/entry/emdb/EMD-9345 | Electron Microscopy Data Bank, EMD-9345 |
| Sobti M, Stewart AG | 2019 | E. coli ATP synthase after incubation with ATP - State B | http://www.ebi.ac.uk/pdbe/entry/emdb/EMD-9346 | Electron Microscopy Data Bank, EMD-9346 |

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
