## [Decision Letter]

Thank you for submitting your article "Cryo-EM reveals distinct conformations of *E. coli* ATP synthase on exposure to ATP" for consideration by *eLife*. Your article has been reviewed by three peer reviewers, including Werner Kühlbrandt as the Reviewing Editor and Reviewer 1, and the evaluation has been overseen by Richard Aldrich as the Senior Editor. The following individuals involved in review of your submission have agreed to reveal their identity: Thomas Meier (Reviewer #2); Christoph Gerle (Reviewer #3).

The reviewers have discussed the reviews with one another and the Reviewing Editor has drafted this decision to help you prepare a revised submission.

Summary:

The manuscript by Sobti et al. is a very useful follow up on the 2016 *eLife* paper mostly by the same authors, which reported the complete cryo-EM structure of the *Escherichia coli* ATP synthase (Sobti et al., 2016). The previous structure showed the epsilon subunit engaging with the α, β and γ subunits preventing rotation and locking the enzyme. In the present research advance, the authors show a new conformation that involves the epsilon C-terminal domain transitioning via an intermediate "half-up" to a condensed "down" state explaining different conformational states of the activated ATPase upon ATP addition. The data are put nicely into a mechanistic model of F1 activation (Figure 4). The new work complements previous work and provides insights into the structural alterations upon ATP binding. The method used to trap this structural alteration was particularly clever: the enzyme was incubated for a given time with an excess of ATP (10 mM), allowing ATP hydrolysis to proceed before plunge freezing. The exact ATP/ADP ratio was also determined by ATP and ADP quantification. Overall, the findings are important for understanding bacterial ATP synthase regulation and bacterial physiology in general.

Essential revisions:

1) It is unclear how we know that the "half-up" conformation is not just an artefact? The ATP synthase used was cysteine-free. Please check whether any cysteine(s) might play a role in the stabilising the "up" or "down" conformations.

2) Ishmukhametov et al., 2010, demonstrated that detergent-solubilised *E. coli* ATPase loses its hydrolysis activity within 4 hours. Can the authors exclude that some inactivation has already taken place when they froze their samples? A few more details e.g. on the timing of freezing/thawing the enzyme in the Results or Materials and methods would be interesting.

3) The *K_d_* of ATP binding to epsilon was 22 mM (the authors mention and discuss this), but why did they freeze the sample in 9.75 mM ATP? Could this partly explain why only a fraction of the observed conformations was "half-up", and the "fully up" conformation was not observed? Maybe there was just not enough ATP available, or the ATP/ADP ratio was not ideal for binding? What happens in the presence of more ATP or a higher ATP/ADP ratio? Presumably 9.75 mM ATP was not the only concentration tested, so it would be interesting to know what happened at higher ATP levels.

4) The map resolution remains significantly below that of other cryo-EM structures of closely related bacterial and eukaryotic ATP synthases. The reasons for this are not clear and an explanation should be at least attempted.

5) Another point that merits more discussion is the partial presence of bound ATP at the epsilon subunit. What is the role of this nucleotide in ATP synthase inhibition? Is it part of a product inhibition mechanism?

6) To convince the sceptical reader that the structure is indeed in the catalytically active state and not in the ADP Mg^2+^-inhibited state, some additional information would help. Please state the source of the chemicals used, especially ATP and the ATPase concentration in mg/ml (as is usual in the cryo-EM field), rather than as a molar concentration. It is also important to know the ADP concentration for an estimate of how many particles were in the ADP Mg-inhibited state at the point when ATP was added. Please add a classification flow chart that clearly indicates the particle numbers at each stage of classification and refinement.

7) Three different rotational states were resolved, but why no substates? Please discuss with respect to single molecule studies. Are the unresolved substates the main limitation on resolution?

8) A key issue in the field is where torsional energy is transiently stored during rotary catalysis. Junge and colleagues have proposed the foot of the central stalk, but cryo-EM structures of inhibited complexes indicate that the peripheral stalk is more flexible. Is this also the case for the *E. coli* complex under working conditions?

[Editors' note: further revisions were requested prior to acceptance, as described below.]

Thank you for submitting your revised research advance "Cryo-EM reveals distinct conformations of *E. coli* ATP synthase on exposure to ATP" to *eLife*. You have dealt with the reviewer's comments very well, and everything is fine except for one small (but important) point:

In your response to the reviewers' comments 7 and 8 about rotational states and stalk flexibility, you write that the peripheral stalk flexes between the three rotary states you resolve, as it does in chloroplasts. You show this in a new figure, which is a great addition to the manuscript.

In your response to point 8 you write that the peripheral stalk flexes as a whole (as expected, and as it does in our structure of the chloroplast ATP synthase), but you did not observe the delta/OSCP hinge that we recently discovered in *Polytomella*. This is not surprising, because the hinge movement is required only between rotary substates, which you do not resolve. In the three primary rotary states, which you do resolve, delta/OSCP is necessarily in the same conformation, and there is no hinge movement. This is also the case in the chloroplast ATP synthase, where rotary substates are likewise not observed, because it is autoinhibited.

It would be good if you could address and clarify this point in a lightly revised manuscript.

---

## [Author Response]

Essential revisions:1) It is unclear how we know that the "half-up" conformation is not just an artefact? The ATP synthase used was cysteine-free. Please check whether any cysteine(s) might play a role in the stabilising the "up" or "down" conformations.

We have modified the second paragraph of the Discussion to discuss this point in detail and a figure (Figure 1—figure supplement 10) has been added to the manuscript to describe the location and distance of the cysteine mutants to the εCTH1. Although we cannot rule out this possibility completely, it is unlikely that the “half-up” conformation we observe here is an artefact generated by cysteine free construct used in this study, because, in our previous study (Sobti et al., 2016), the same cysteine-free enzyme used in the present work did not show the “half-up” state. In addition, previous studies using γC87S mutant protein did not show and change in ε inhibition (Duncan et al., 1995). Moreover, in the structure of the native enzyme (Figure 1—figure supplement 10), loops in the γ subunit separate the mutated cysteines and residues in the εCTH1, so that they are not in direct contact with the εCTD.

2) Ishmukhametov et al., 2010, demonstrated that detergent-solubilised *E. coli* ATPase loses its hydrolysis activity within 4 hours. Can the authors exclude that some inactivation has already taken place when they froze their samples? A few more details e.g. on the timing of freezing/thawing the enzyme in the Results or Materials and methods would be interesting.

The text has been expanded, with an extra paragraph in the Discussion (seventh paragraph), to address this point. The enzyme that was used in this study was made as promptly as possible. The purification method contained two changes compared to Ishmukhametov et al., 2010: (i) digitonin was used as the detergent and (ii) a size exclusion step was included. To test whether the enzyme retained hydrolysis activity over 4 hours at room temperature, the same regeneration assays were performed at the same time points as in Ishmukhametov et al., 2010 (Figure 1—figure supplement 12). Although a small amount of activity is lost over 8 hours, the reduction in ATPase activity is far less than seen previously, suggesting that either digitonin is keeping the protein more stable, or the size exclusion chromatography is removing aggregation nucleates.

3) The K_d_ of ATP binding to epsilon was 22 mM (the authors mention and discuss this), but why did they freeze the sample in 9.75 mM ATP? Could this partly explain why only a fraction of the observed conformations was "half-up", and the "fully up" conformation was not observed? Maybe there was just not enough ATP available, or the ATP/ADP ratio was not ideal for binding? What happens in the presence of more ATP or a higher ATP/ADP ratio? Presumably 9.75 mM ATP was not the only concentration tested, so it would be interesting to know what happened at higher ATP levels.

The text has been expanded (subsection “Structure of *E. coli* F_1_F_o_ ATP synthase in the presence of ATP”, first paragraph and Discussion, fifth paragraph) to include our rationale for using 9.75 mM ATP and 0.3 mM ADP while freezing. In short, this was to simulate nucleotide concentrations found in *E. coli* and so identify conformations likely to be present in vivo. Importantly, autoinhibition appears to be removed at 1 mM ATP, well below the concentration we employed, and the balance of evidence indicates that nucleotide binding to the ε subunit is unlikely to make a major contribution to the physiological regulation of the ATP synthase in *E. coli*. Many biochemical studies of isolated *E. coli* F_1_ have shown that ε inhibition is non-competitive *vs* ATP^3^, and high ATP concentrations do not activate the ATPase of wild-type membranes compared to membranes lacking the entire εCTD (Nakanishi-Matsui et al., 2014). Because weak density attributable to ATP bound to εCTH1 was only seen with the down conformation, it is possible that the half-up conformation, with the associated probable disorder of εCTH2, might result when ATP is not bound to the ε subunit. We mention this possibility in the fourth paragraph of the Discussion and highlight that it would need additional work to evaluate. However, this possibility is peripheral to the main thrust of the present manuscript and would be a major investigation on its own. Imaging in higher or lower concentrations of ATP/ADP is something we are actively working on, but is well beyond the scope of this manuscript and using much higher ATP concentrations is not straight forward (e.g. due to large changes in ionic strength).

4) The map resolution remains significantly below that of other cryo-EM structures of closely related bacterial and eukaryotic ATP synthases. The reasons for this are not clear and an explanation should be at least attempted.

To improve the resolution of the maps, additional micrographs were taken and merged with the previous data presented. This increased the overall resolution to ~5 Å (from the ~6 Å previously achieved), with a small number of side chains now visible in the centre of the F_1_ motor. Although this resolution is still far below that of other structures published, the new maps do not change the conclusions drawn in the manuscript which are robust and do not require fine detail. The Discussion has been updated to include possible reasons for the moderate resolution observed (accelerator voltage, ice thickness, detergent, etc, together with multiple rotational states as discussed in point 7).

5) Another point that merits more discussion is the partial presence of bound ATP at the epsilon subunit. What is the role of this nucleotide in ATP synthase inhibition? Is it part of a product inhibition mechanism?

Although the literature is not completely clear about this question, the balance of evidence indicates that nucleotide binding to the ε subunit is unlikely to make a major contribution to the physiological regulation of the ATP synthase in *E. coli*. Many biochemical studies of isolated *E. coli* F_1_ have shown that ε inhibition is non-competitive *vs* ATP (Sielaff, Duncan and Borsch, 2018) and high ATP concentrations do not activate the ATPase of membranes containing wild-type protein compared to membranes lacking the entire εCTD (Nakanishi-Matsui et al., 2014). Also, no preference has been seen between ATP and ADP in the isolated *E. coli* epsilon subunit (Sielaff, Duncan and Borsch, 2018) and mutagenesis of this region in related organisms does not appear to interfere with the mechanism (Ferguson et al., 2016). We have discussed this point in greater detail in the Discussion.

6) To convince the sceptical reader that the structure is indeed in the catalytically active state and not in the ADP Mg^2+^-inhibited state, some additional information would help. Please state the source of the chemicals used, especially ATP and the ATPase concentration in mg/ml (as is usual in the cryo-EM field), rather than as a molar concentration. It is also important to know the ADP concentration for an estimate of how many particles were in the ADP Mg-inhibited state at the point when ATP was added. Please add a classification flow chart that clearly indicates the particle numbers at each stage of classification and refinement.

The source and product code of the ATP used has been added to the Materials and methods. Concentrations have been quoted in “mg/ml” as well as molar concentrations. The ADP concentration in 10 mM ATP was calculated to be 0.05 mM (n=3, stdev=0.002) as determined by HPLC and is now quoted in the text, in hindsight this should have been included. A flowchart has been added to the manuscript describing the particle numbers at each stage of classification and refinement (Figure 1—figure supplement 6). We have not commented on the proportion of particles in each state, because we felt that this would be over interpreting the data (i.e. how do we know that we have sorted all the particles correctly? Could we be preferentially imaging/picking a particular state?)

7) Three different rotational states were resolved, but why no substates? Please discuss with respect to single molecule studies. Are the unresolved substates the main limitation on resolution?

There are likely other rotational sub-states present in our data, and this may impact our resolution. We had tried similar methods employed by others to classify additional sub-states (e.g. classification after 3D refinement) and we attempted this again with the larger number of images/particles present in the updated data. However, we were unable to classify additional rotational states. This could be due to the very small proportion of the time spent in some of these states while undergoing rotation. For example, even with a large nanorod attached to the c-ring, rotational sub-states were only overserved in a viscous solution containing 30% PEG using single molecule methods (Ishmukhametov et al., 2010). This has been discussed further in the Discussion and compared and contrasted to the new cryo-EM study of *Polytomella sp.* which contains 13 rotational sub-states.

8) A key issue in the field is where torsional energy is transiently stored during rotary catalysis. Junge and colleagues have proposed the foot of the central stalk, but cryo-EM structures of inhibited complexes indicate that the peripheral stalk is more flexible. Is this also the case for the *E. coli* complex under working conditions?

We have expanded the Discussion to address this point. A recent study on *Polytomella sp.* mt. ATP synthase has shown a single hinge in the peripheral stalk to be essential for coupled rotation of ATP synthase. To investigate whether this same hinge is present in *E. coli* ATP synthase under working conditions, we subtracted the peripheral stalk and supposed the three rotational states on either the membrane domain or the N-terminal domain of subunit delta (to highlight the same possible hinge region seen in *Polytomella sp.*) (Figure 3—figure supplement 2:). Interestingly, the flexibility of the peripheral stalk appears to come from the entire stalk, rather than just a hinge in delta (OSCP in *Polytomella sp.* mt. ATP synthase). This is now discussed in the Discussion section.

[Editors' note: further revisions were requested prior to acceptance, as described below.]

Thank you for submitting your revised research advance "Cryo-EM reveals distinct conformations of *E. coli* ATP synthase on exposure to ATP" to eLife. You have dealt with the reviewer's comments very well, and everything is fine except for one small (but important) point:In your response to the reviewers' comments 7 and 8 about rotational states and stalk flexibility, you write that the peripheral stalk flexes between the three rotary states you resolve, as it does in chloroplasts. You show this in a new figure, which is a great addition to the manuscript.

Figure 3—figure supplement 2 has now been updated with additional text:

“Black arrows show the relative movement between the C-terminal domain of subunit δ and stator of F_o_, which is likely mediated by flection and twisting in the peripheral stalk.”

In your response to point 8 you write that the peripheral stalk flexes as a whole (as expected, and as it does in our structure of the chloroplast ATP synthase), but you did not observe the delta/OSCP hinge that we recently discovered in Polytomella. This is not surprising, because the hinge movement is required only between rotary substates, which you do not resolve. In the three primary rotary states, which you do resolve, delta/OSCP is necessarily in the same conformation, and there is no hinge movement. This is also the case in the chloroplast ATP synthase, where rotary substates are likewise not observed, because it is autoinhibited.

We have edited the Discussion to reflect this. It now reads:

“Recent work on *Polytomella sp*. mitochondrial ATP synthase (Murphy et al., 2019) has described this ATP synthase in great detail, showing a flexible coupling between the F_1_ and F_o_ motors mediated by a hinge region in *OSCP* (analogous to the δ subunit in *E. coli* ATP synthase). […] We do not observe movement in the delta/OSCP hinge, most likely due to our inability to resolve rotary substates in our maps.” With the chloroplast ATP synthase structure now cited here as well.